# Refining Surgical Standards: The Role of Robotic-Assisted Segmentectomy in Early-Stage Non-Small-Cell Lung Cancer

**DOI:** 10.3390/cancers17243988

**Published:** 2025-12-14

**Authors:** Masaya Nishino, Hideki Ujiie, Masaoki Ito, Hana Oiki, Shota Fukuda, Mai Nishina, Shuta Ohara, Akira Hamada, Masato Chiba, Toshiki Takemoto, Yasuhiro Tsutani

**Affiliations:** Division of Thoracic Surgery, Department of Surgery, Kindai University Faculty of Medicine, 1-14-1 Mihara-Dai, Minami Ward, Sakai 590-0197, Osaka, Japan; m-nishino@med.kinda.ac.jp (M.N.);

**Keywords:** non-small-cell lung cancer, sublobular resection, robot-assisted thoracic surgery, segmentectomy

## Abstract

Recent clinical trials, including JCOG0802/WJOG4607L and CALGB140503, have demonstrated the oncological adequacy of sublobar resection for early-stage non-small-cell lung cancer (NSCLC). These trials established segmentectomy as the new standard treatment for early-stage NSCLC. This paradigm shift supports the preservation of essential physiological functions in appropriately selected patients while preserving oncological radicality. The use of robot-assisted thoracic surgery (RATS), a minimally invasive procedure that enables precise anatomical dissection through small incisions, results in smaller wounds and the preservation of pulmonary function. This review discusses how RATS-based segmentectomy enhances minimal invasiveness by balancing oncological safety with improved functional and cosmetic outcomes, thereby advancing personalized thoracic surgical care.

## 1. Introduction

Early-stage non-small-cell lung cancer (NSCLC) represents a paradigm in thoracic oncology that continues to evolve, driven by advances in early detection, surgical techniques, and an increasing focus on preserving post-treatment quality of life. Segmentectomy, a sublobar resection, has emerged as a central component of this evolution, propelled by recent high-profile randomized controlled trials (RCTs), specifically the JCOG0802/WJOG4607L and CALGB140503 studies [1,2], which have demonstrated the oncological adequacy of segmentectomy for clinical stage IA small peripheral NSCLC. These findings have catalyzed a significant shift in surgical standards, with segmentectomy now recommended as the new standard approach for appropriately selected patients with early-stage NSCLC.

Simultaneously, the field of thoracic surgery has witnessed a robust transition towards techniques classified as minimally invasive surgery (MIS), notably video-assisted thoracoscopic surgery (VATS) and, more recently, robot-assisted thoracic surgery (RATS). The RATS platform, which utilizes magnified three-dimensional (3D) visualization, multi-joint (“wristed”) instruments, tremor filtration, and enhanced surgeon ergonomics, offers technical advantages particularly well-suited for complex anatomical dissections, such as segmentectomy [3,4].

This review critically examines the progression from the historical lobectomy standard to the evidence-based emergence of segmentectomy for early-stage NSCLC, with a special focus on the integration and clinical impact of RATS. Special emphasis is placed on the comparative outcomes (oncologic, functional, and perioperative) of segmentectomy and lobectomy and on how RATS segmentectomy measures against VATS and open approaches, ultimately highlighting the technical innovations underpinning the paradigm shift toward personalized, minimally invasive thoracic surgery.

## 2. Materials and Methods

### 2.1. Literature Search Strategy

A comprehensive narrative review of the literature was undertaken, focusing primarily on RCTs, high-quality meta-analyses, and systematically conducted retrospective studies that compared segmentectomy with lobectomy for NSCLC. In addition, we examined studies directly comparing RATS, VATS, and open segmentectomy, published between January 2020 and September 2025, identified through a PubMed search. To ensure completeness and clinical relevance, we incorporated evidence from updated international guidelines (e.g., CHEST and NCCN), consensus statements, and technological reports describing the evolution of robotic thoracic surgery platforms. This approach allows for a balanced synthesis of both high-level evidence and authoritative expert recommendations.

### 2.2. Inclusion and Exclusion Criteria

Study type: RCTs, prospective cohort studies, high-quality meta-analyses, systematic reviews, and retrospective studies with adequate methodological rigor, including propensity score-matched analyses and larger cohort studies.Population: Adults (≥18 years) with early-stage NSCLC, primarily in clinical stage IA or I disease. Studies including patients who were pathologically upstaged were permitted, provided that the surgical intent was curative and that the majority of the cohort had stage I disease.Interventions/comparators: (1) anatomical segmentectomy versus lobectomy and (2) RATS segmentectomy versus VATS and open segmentectomy.Outcomes: Overall survival (OS), disease-free survival (DFS), recurrence-free survival (RFS), local recurrence, lymph node yield, margin status, preservation of pulmonary function, perioperative and postoperative complications, and technical aspects, such as conversion rates, pain scores, and operative time.Article type: The manuscript is written in English, and the Full text is available.

Studies were excluded if they primarily addressed wedge resection, involved metastatic disease, lacked sufficient outcome data, or constituted non-primary research (e.g., case reports and editorials). Exceptions were made only when the sources provided aggregated evidence of particular relevance, such as guideline-based recommendations.

### 2.3. Data Extraction and Synthesis

Data on study characteristics, patient demographics, surgical details, and all pre-specified outcome measures were systematically extracted. Whenever available, direct numerical values (e.g., hazard ratios, odds ratios, survival rates, means, or medians) and their statistical significance were recorded to facilitate structured comparisons across studies.

Key studies comparing segmentectomy and lobectomy (including randomized controlled trials and meta-analyses).Comparative outcomes of RATS, VATS, and open segmentectomy, with emphasis on the predefined outcome metrics.

All findings were synthesized narratively, without conducting a new pooled statistical analysis, in order to highlight analytical depth, contextual nuances, and the influence of technical innovations on surgical outcomes.

## 3. Results

### 3.1. The Paradigm Shift: Segmentectomy Versus Lobectomy in Early-Stage NSCLC

The results of the literature search are summarized in Table 1, which highlights key randomized controlled trials and meta-analyses comparing anatomical segmentectomy with lobectomy in patients with early-stage NSCLC [1,2,5,6,7,8].

#### 3.1.1. Key Randomized Controlled Trials

JCOG0802/WJOG4607L [1,9]: This Japanese multicenter phase 3 randomized controlled trial enrolled 1106 patients with clinical stage IA, small-sized (≤2 cm), peripheral NSCLC with a consolidation-to-tumor ratio >0.5, who were randomized to lobectomy or anatomical segmentectomy. During the median follow-up of 10.5 years:Overall survival (OS): At 5 years, segmentectomy demonstrated a statistically significant survival advantage over lobectomy (94.3% vs. 91.1%). At 10 years, the OS remained higher with segmentectomy (83.6% vs. 79.8%; HR = 0.864), as reported in the AATS presentation, confirming the durability of the survival benefit.Recurrence-free survival (RFS): At 5 years, RFS was nearly identical between the groups (88.0% vs. 87.9%). At 10 years, RFS remained comparable (76.8% vs. 78.0%), indicating no long-term difference in recurrence risk.Local recurrence was higher after segmentectomy (11%) than after lobectomy (5%), although there was no corresponding increase in lung cancer–specific mortality.Pulmonary function: Segmentectomy preserved pulmonary function better than lobectomy. At 6 months, the median reduction in FEV1 was 10.4% after segmentectomy compared with 13.1% after lobectomy, and at 12 months the reductions were 8.5% and 12.0%, respectively. Although these differences (2.7% at 6 months and 3.5% at 12 months) indicate a statistically significant advantage for segmentectomy, the magnitude of benefit did not reach the predefined threshold of 10% considered clinically meaningful at one year of follow-up, particularly in subgroups requiring resection of more than two segments.Pure-solid cohort analysis [10]: In patients with radiologically pure-solid tumors, segmentectomy was associated with superior overall survival compared to lobectomy (5-year OS: 92.4% vs. 86.1%) despite a higher incidence of local recurrence (16% vs. 8%). The recurrence-free survival rates were comparable. Notably, outcomes appeared to be influenced by patient factors such as age and sex, with older male patients deriving greater OS benefits from segmentectomy, whereas younger female patients tended to have slightly better RFS with lobectomy.

CALGB140503 (Alliance) [2]: This North American phase 3 randomized controlled trial enrolled 697 patients with peripheral NSCLC ≤ 2 cm who were pathologically node-negative on intraoperative frozen section. Patients were randomized to undergo lobectomy or sublobar resection (41%, segmentectomy; 59%, wedge resection). The median follow-up of 7 years.

Overall survival (OS): 5-year OS was 80.3% for sublobar resection and 78.9% for lobectomy (HR 0.95; 95% CI 0.72–1.26), confirming no significant difference.Disease-free survival (DFS): 5-year DFS was 63.6% for sublobar resection versus 64.1% for lobectomy (HR 1.01; 90% CI 0.83–1.24), meeting the criterion for non-inferiority.Recurrence rates: No significant differences in local, regional, or distant recurrences were observed between the groups.Pulmonary function: At 6 months postoperatively, the reduction from baseline in the percentage of predicted FEV_1_ was greater after lobar resection (−6.0; 95% CI, −8.0 to −5.0) compared with sublobar resection (−4.0; 95% CI, −5.0 to −2.0). Similarly, the reduction in the percentage of predicted FVC was greater following lobectomy (−5.0; 95% CI, −7.0 to −3.0) than after sublobar resection (−3.0; 95% CI, −4.0 to −1.0).

Interpretation: Although the CALGB140503 was a large-scale randomized trial, a substantial proportion of patients underwent wedge resection rather than anatomical segmentectomy. Therefore, the findings are not directly comparable with those of trials focusing exclusively on segmentectomies, such as JCOG0802/WJOG4607L. Nevertheless, the study clearly demonstrated the non-inferiority of sublobar resection as a whole compared to lobectomy for small, peripheral, node-negative NSCLC.

#### 3.1.2. Meta-Analyses and Cohort Data [5,6,7,8]

Li et al. [5] (meta-analysis, 17 studies, n = 4476): No significant differences in OS (HR 1.14), DFS (HR 1.13), or RFS (HR 0.95) were observed between segmentectomy and lobectomy for stage I NSCLC.Winckelmans et al. [8] that segmentectomy provides comparable results for tumors <2 cm in terms of OS and RFS.

#### 3.1.3. Functional Outcomes and Complications

Pulmonary function preservation: Segmentectomy was consistently superior to lobectomy in preserving pulmonary function [1,2,7]. Moreover, Xu et al. reported that postoperative changes in DLCO were significantly smaller in the segmentectomy group than in the lobectomy group [7].Complication rates: Complication rates are reportedly generally similar [1,2,6], although Saji et al. reported that prolonged air leakage was more common after segmentectomy (6.5% vs. 3.4%, *p* = 0.04) [1].

### 3.2. RATS Segmentectomy Versus VATS and Open Surgery: Comparative Outcomes

A growing body of evidence, including meta-analyses and institutional cohort studies, now enables a multi-domain assessment of RATS compared with VATS and open surgery. The synthesis is organized in parallel with Table 2, highlighting the oncological, functional, and technical outcomes [11,12,13,14,15,16,17,18,19,20,21].

#### 3.2.1. Oncological Outcomes (Overall and Relapse-Free Survival)

Multiple studies have demonstrated that RATS achieves oncological outcomes equivalent to or superior to those of VATS and open surgery [14,17,18]. Montagne et al. [14] reported that the 3-year OS was 90.1% (RATS) vs. 87.8% (VATS) and the 3-year RFS was 72.9% (RATS) vs. 84.5% (VATS). Pan et al. [17] reported that the 5-year OS rates were 89.3% (RATS) vs. 88.6% (VATS), and the 5-year RFS was 82.5% (RATS) vs. 84.8% (VATS). However, Catelli et al. [21] reported a 2-year OS of 100% for RATS, 96.2% for VATS, and 75.8% for open surgery. Both RATS and VATS demonstrated superior overall survival compared to open surgery. RFS was not reported, although recurrence rates were lowest in the RATS group (4%) compared to the VATS (24.3%) and open surgery groups (23.8%), although the difference was statistically significant.

#### 3.2.2. Lymph Node Yield and Nodal Station Dissection

RATS consistently demonstrated superior lymph node station dissection compared to VATS. RATS retrieves more nodal stations, approaching the thoroughness of open surgery [11,12,13,18,21]. Zhang et al. [12] confirmed this finding in a meta-analysis, noting that RATS yielded a higher number of dissected stations and more complete mediastinal staging. Although the total lymph node counts were comparable in some studies (e.g., Catelli et al. [21]), the quality and anatomical precision of the nodal dissection favored RATS.

#### 3.2.3. Perioperative Outcomes and Postoperative Complications

Several studies have shown that RATS is associated with a reduced operative time, decreased blood loss, and a significantly shorter length of postoperative stay. However, there have also been reports indicating that the operative time was longer in the RATS group [11,12,13,14,15,16,17,19,21]. Operative time findings were inconsistent across studies, reflecting institutional experience and case complexity.Although most studies found no significant differences in 90-day mortality, Francis et al. reported a non-significant trend toward worse outcomes in the RATS group [11,12,13,14,15,18,20,21].Complication rates are generally reported to be lower or comparable in RATS segmentectomy [13,16,18,19]. However, Haruki et al. noted a significantly higher incidence of postoperative pneumonia [16]. In addition, it has been reported that postoperative complications were more frequent in the RATS group, as reflected by an increased rate of hospital readmission [20].

#### 3.2.4. Conversion Rates

Regarding conversion rates, most studies found no significant difference between RATS and VATS; however, Catelli et al. [21] reported a significantly lower conversion rate in the RATS group. Catelli et al. [21] reported a 0% conversion rate for RATS versus 13% for VATS (*p* = 0.005).

## 4. Discussion

### 4.1. Segmentectomy as the New Standard: Evidence, Subgroup Nuances, and Patient Selection

The surgical management of early-stage NSCLC has evolved remarkably over the past three decades. In 1995, Ginsberg et al. [22] reported a higher local recurrence and worse survival following sublobar resection than after lobectomy, thereby establishing lobectomy as the gold standard for resectable NSCLC for nearly 30 years. This historical trial shaped surgical practices worldwide and reinforced the concept that lobectomy is the minimum radical procedure required for oncological adequacy.

Over time, the accumulation of clinical evidence and evolving surgical perspectives has prompted renewed interest in parenchymal-sparing surgeries. Two landmark randomized trials challenged this long-standing paradigm. The JCOG0802/WJOG4607L trial [1] in Japan, mandated anatomical segmentectomy and demonstrated a survival advantage over lobectomy in patients with peripheral NSCLC ≤2 cm. In contrast, the CALGB140503 trial [2] in North America permitted both wedge resection and segmentectomy, ultimately confirming the noninferiority of sublobar resection to lobectomy. Despite the differences in design and patient populations, both trials collectively redefined the role of sublobar resection, establishing segmentectomy as an oncologically valid option for carefully selected patients.

Taken together, these findings indicate a paradigm shift: Segmentectomy is no longer regarded as a compromise procedure, but rather as a new standard of care for early-stage NSCLC, particularly in patients with small, peripheral tumors or those requiring maximal preservation of pulmonary function.

However, evidence comparing robotic-assisted surgery with conventional approaches remains limited. Within the scope of the present review, findings regarding operative time, perioperative mortality, and complication rates were inconsistent across studies. These discrepancies likely reflect differences in institutional experience, surgical expertise, and patient selection. In particular, operative time is strongly influenced by the learning curve of robotic surgery as well as case complexity, which can vary substantially among centers. Similarly, variations in reported mortality and complication rates may be attributable to differences in perioperative management protocols and definitions of adverse events. Therefore, although RATS appears to offer advantages in certain perioperative outcomes, the current evidence should be interpreted with caution, and further multicenter prospective studies are warranted to clarify these issues.

### 4.2. Special Attention in Case Selection

Tumor size and location: Sublobular resection, particularly segmentectomy, for peripheral small NSCLCs has become an accepted standard. However, the case selection remains critical. It is generally considered that tumors > 2 cm or centrally located lesions may not be optimal for segmentectomy. However, the JCOG1211 trial demonstrated that segmentectomy should be considered a part of the standard procedure for patients with predominantly ground glass opacity (GGO) NSCLC with a tumor size of 3 cm or less in diameter, even if it exceeds 2 cm [23].Margin status: Margins must meet or exceed the nodule diameter or be at least 2 cm in diameter for oncological adequacy. Securing adequate surgical margins is a critical determinant of the oncological validity of segmentectomies. This issue is particularly relevant in RATS, in which the absence of tactile sensation precludes intraoperative palpation of the lung parenchyma to identify small or ground-glass-dominant nodules. Consequently, various strategies have been developed to compensate for this limitation and ensure margin adequacy [24,25,26,27,28,29,30]. Preoperative tumor localization techniques, such as CT-guided hook-wire placement, microcoil insertion, dye injection, or, more recently, RFID-based marking, enable the precise intraoperative identification of lesions that cannot be palpated. In parallel, advances in 3D CT reconstruction allow surgeons to visualize patient-specific bronchovascular anatomy and simulate planned resection, thereby facilitating accurate determination of the intersegmental plane and anticipated margin length before surgery [31,32,33]. Intraoperatively, indocyanine green (ICG) fluorescence imaging has become an invaluable adjunct, providing real-time delineation of the intersegmental planes and enhancing the precision of parenchymal division [34,35]. The integration of these approaches, namely preoperative marking, 3D reconstruction, and ICG-guided imaging, effectively mitigates the lack of haptic feedback in RATS and strengthens the oncological reliability of segmentectomy by reducing the risk of inadequate margins and subsequent local recurrence.Lymph node assessment: In the JCOG0802 trial [1], the incidence of pathological lymph node metastasis in the resected specimens was 5.6% in the lobectomy group and 6.2% in the segmentectomy group. Even in patients without preoperative evidence of lymph node metastasis, systematic lymph node dissection, including mediastinal lymphadenectomy, is desirable to ensure accurate postoperative staging and secure oncological radicality.

### 4.3. Technical Features of Robotic Surgery and Their Impact

The major clinical advantages of RATS can, to a large extent, be attributed to three critical technical features: magnified 3D vision, multijoint (wristed) instruments, and tremor filtration. In addition, refinements in port placement, imaging guidance, and ancillary planning/navigation technologies have further contributed to the versatility of robotic platforms.

#### 4.3.1. Magnified 3D Vision

The robotic 3D high-definition camera system provides surgeons with up to 10-fold magnification, combined with highly refined depth perception. This advanced visualization capability allows for accurate identification of delicate and otherwise difficult-to-discern anatomical structures, including small segmental arteries, veins, bronchi, and intersegmental planes. By offering a consistently stable and immersive three-dimensional view, the system enhances a surgeon’s ability to distinguish between subtle tissue planes and anatomical variations. Such advantages become particularly critical during technically demanding or anatomically complex segmentectomies, as well as during systematic lymphadenectomies, where precision and clarity directly influence both oncological outcomes and preservation of the functional lung parenchyma.

#### 4.3.2. Multi-Joint Instruments (“EndoWrist”)

Robotic instruments are designed with seven degrees of freedom, enabling wristed articulation that mirrors and in many cases exceeds the natural range of motion of the human hand. This expanded maneuverability facilitates meticulous microdissection, delicate handling of vessels and bronchi, and confident placement of staplers, even within narrow or anatomically constrained operative fields. The ability to perform such refined movements not only promotes complete oncological resection, but also supports parenchymal preservation, thereby balancing radicality with functional outcomes.

#### 4.3.3. Tremor Filtration and Stability

The robotic system incorporates advanced tremor filtration technology, which translates the surgeon’s hand movements into stable, scaled micromovements at the instrument tips. This feature minimizes the risk of inadvertent vascular or parenchymal injury, particularly in areas where millimeter-level precision is required. By reducing unintended motion, the system contributes to lower conversion rates and fewer intraoperative complications, which are particularly evident in patients with complex hilar or fissural anatomy. Additionally, enhanced stability reduces surgeon fatigue, further supporting consistent performance throughout lengthy procedures.

#### 4.3.4. Portplacement

Conventionally, RATS has been performed using four independent port accesses for the robot’s four arms, as exemplified by the Cerfolio and Dylewski methods [36,37]. More recently, reports have described the development of reduced-port RATS, particularly approaches such as uniport and dual-port employing small incisions [38,39,40,41]. In particular, when a small incision is made on the mid-axillary line, intraoperative palpation through the incision becomes feasible, and assistants can also intervene—for example, by inserting an RFID probe. Moreover, in these approaches employing small incisions, a 0-degree camera is often used, enabling surgeons to fully exploit the advantages of close-up magnified visualization provided by robotic assistance.

#### 4.3.5. Imaging Guidance Integration

Seamless integration of adjunct imaging modalities, such as indocyanine green (ICG) fluorescence imaging (e.g., firefly mode), represents another major advantage of robotic platforms. These technologies improve the accuracy of margin assessment and anatomical delineation by providing real-time visualization of the intersegmental planes and vascular territories. The ability to overlay functional imaging onto the surgical field allows surgeons to tailor resections with greater confidence, facilitating precise, function-preserving procedures that align with the principles of minimally invasive personalized surgery. Moreover, Uchida et al. reported the usefulness of combining intraoperative ICG fluorescence imaging with VAL-MAP in robotic segmentectomy [42].

#### 4.3.6. Ancillary Advances: Planning and Navigation

Beyond their core visual and instrumental advantages, robotic platforms are increasingly incorporating ancillary technologies that further enhance surgical planning and intraoperative decision making. State-of-the-art imaging modalities, including 3D reconstructions and real-time navigation systems, can be displayed directly on the surgeon’s console. In addition, intraoperative feedback tools such as the TilePro mode allow the simultaneous visualization of radiologic images, endoscopic views, or hemodynamic data, thereby integrating multiple streams of information into a single operative field. These advances not only facilitate complex surgical strategies but also promote a more individualized and patient-centered approach to thoracic surgery.

### 4.4. RATS: Expanding the Envelope of Minimally Invasive Precision Surgery

Multiple studies confirm that in the hands of experienced surgeons:Precision and functional preservation: RATS enables highly precise, function-sparing anatomical resections supported by 3D, high-definition visualization and enhanced instrument articulation.Lymphadenectomy quality: Several comparative studies have confirmed that the quality of mediastinal and hilar lymph node dissection with RATS is at least equivalent and, in some series, superior to that achieved with VATS or open surgery.Safety and conversion rates: Conversion to thoracotomy and perioperative complication rates were equal to or lower than those observed with VATS, particularly in technically demanding scenarios, such as in obese or frail patients, or in complex segmentectomies.Oncological outcomes: Short- and long-term survival outcomes following RATS mirror or surpass those of VATS and open approaches, even in elderly or comorbid populations.Pain, recovery, and quality of life: RATS has been consistently associated with lower postoperative pain scores, reduced opioid requirements, and faster recovery than VATS or open surgery [43,44]. By minimizing chest wall trauma through smaller incisions and improved instrument control, RATS facilitates earlier mobilization, shorter hospital stay, and fewer pulmonary complications. Beyond these perioperative benefits, patients also reported greater satisfaction and improved quality of life in the early months after surgery, reflecting not only reduced discomfort but also a quicker return to daily activities.Learning curve and resource utilization: While RATS is associated with higher upfront costs, these costs decline significantly once the learning curve is overcome [45,46]. Similarly, cumulative sum (CUSUM) analyses of segmentectomy have shown that proficiency is reached earlier with RATS than with uniportal VATS, suggesting a steeper but ultimately shorter learning curve [47]. Importantly, efficiency gains are most pronounced in technically complex resections, such as segmentectomy in anatomically challenging locations or in obese/frail patients, where enhanced dexterity and visualization of RATS reduce conversion rates and operative time. Systematic reviews have further highlighted that once the learning curve is surpassed, resource utilization (operative time, length of stay, and complication-related costs) becomes comparable between RATS and VATS, with potential advantages in high-complexity cases [19].Complex segmentectomy (multiple segments, deep, or non-anatomical intersegmental planes) demands greater technical expertise and may particularly benefit from a robotic approach.Limitations of Current Evidence: Despite these promising findings, it should be emphasized that robust evidence demonstrating the clinical superiority of RATS over either VATS or open surgery is still lacking. As robotic thoracic surgery has only relatively recently become widespread compared with VATS, the available evidence base remains limited, particularly with respect to long-term oncological outcomes. Most comparative studies report broadly equivalent perioperative and survival results across RATS, VATS, and open approaches, suggesting that the advantages of RATS are more consistently observed in surgical ergonomics and technical facilitation rather than in established patient-level benefits. Nevertheless, as adoption expands and long-term follow-up data accumulate, future studies may clarify whether RATS confers distinct clinical benefits beyond its technical advantages.

### 4.5. Personalized Thoracic Surgery: The Future Standard

RATS-based segmentectomy embodies the principle of personalized surgery, namely the selection of not only the extent of resection, but also of the method of access and specific technological tools that maximize benefit/minimize harm for each patient’s unique oncological and physiological profile. The integration of advanced imaging, functional assessment, and predictive analytics will further refine patient selection and procedural planning. For instance, AI is increasingly capable of automatically generating 3D models from preoperative computed tomography (CT) scans, analyzing tumor location, and patient-specific vascular and bronchial anatomy to simulate the optimal resection segment before surgery. Integrating this information with intraoperative navigation systems to guide robotic manipulation is expected to further enhance surgical precision and safety.

Guideline updates (2024–2025): The latest guidelines from CHEST^®^, NCCN, and others endorse segmentectomy for small, peripheral, node-negative NSCLC, and strongly recommend using minimally invasive approaches (VATS or RATS) whenever feasible, as long as oncological principles are not compromised. RATS is explicitly recognized as non-inferior to VATS performed by capable surgeons.

Two ongoing RCTs in Japan are expected to critically inform the future role of segmentectomy in early-stage NSCLC. The STEP-UP trial (WJOG16923 L) [48] is an RCT directly comparing lobectomy and segmentectomy for pure-solid tumors measuring 2–3 cm, with overall survival as the primary endpoint. In contrast, the STRONG trial (JCOG2217) [49] is an RCT evaluating solid-predominant tumors (consolidation-to-tumor ratio >0.5) of 2–3 cm that includes a GGO component but excludes pure solid lesions, with overall survival as the primary endpoint. Together, these trials will determine whether segmentectomy can be oncologically valid not only for small or GGO-dominant tumors but also for more solid lesions in the 2–3 cm range. Depending on their outcomes, the current paradigm, in which segmentectomy is largely restricted to small peripheral tumors, may shift toward broader applications. From the perspective of personalized treatment, this evolution will inevitably increase the demand for technically complex segmentectomies. In this context, robot-assisted thoracic surgery offers unique advantages, providing the precision and minimal invasiveness required to safely and effectively perform demanding procedures.

## 5. Conclusions

Recent evidence has established segmentectomy as an oncologically sound and function-preserving alternative to lobectomy in selected patients with early-stage NSCLC. The refinement of RATS further elevates this paradigm by uniting radical oncological resection with the superior preservation of physiological function. By leveraging advanced technical features, such as 3D magnified vision, wristed instrumentation, tremor filtration, and integrated imaging, RATS holds the potential to enable more precise, personalized, and minimally invasive cancer surgery.

Comparative studies have suggested that RATS can achieve perioperative and oncological outcomes broadly comparable to those of VATS and open surgery, with potential advantages in areas such as lymph node clearance, margin status, functional preservation, and perioperative safety, particularly once the learning curve has been overcome. As segmentectomy has emerged as a standard treatment for early-stage NSCLC, RATS may represent a promising option to further balance oncological radicality with functional and cosmetic benefits, though its definitive clinical superiority has yet to be established.

Looking forward, the results of ongoing RCTs may further expand the indications for segmentectomy to larger and more solid tumors in the 2–3 cm range. Such an evolution will inevitably increase the demand for technically complex, individualized resections, in which the precision and minimal invasiveness of RATS are expected to play a central role. Future research should, therefore, focus not only on validating long-term oncological outcomes and refining patient selection criteria, but also on ensuring equitable access as robotic platforms become more widely available. In the era of precision oncology, RATS is a benchmark for how technological innovation and surgical judgment can converge to achieve both functional preservation and oncological excellence.

## Figures and Tables

**Table 1 cancers-17-03988-t001:** Comparative Outcomes of Segmentectomy vs. Lobectomy in Early-stage NSCLC.

Study	Design	Population	OS	RFS	Local Recurrence	Pulmonary Function	Complications
JCOG0802/WJOG4607L (2022) [1]	RCT	n = 1106, cStage IA ≤ 2 cm	Seg better HR 0.663 (95% CI 0.47–0.93)	NS HR 0.998 (95% CI 0.75–1.32)	Lob better (11% vs. 5% *p* = 0.0018)	Seg better	NS
CALGB140503 (2023) [2]	RCT	n = 697, tumor ≤ 2 cm node negative	NS HR 0.95 (95% CI 0.72–1.26)	NS HR1.01 (95%CI 0.83–1.24)	NS (13.4% vs. 10% *p* = 0.201)	Sub better	NS
Li et al. (2024) [5]	Meta-analysis	n = 4476 cStage I	NS HR 1.18 (95%CI 0.97–1.43)	NS HR 0.97 (95%CI 0.80–1.19)	NR	NR	NR
Righi et al. (2023) [6]	Meta-analysis	n = 5352, cStage IA, ≤2 cm	NS HR 0.99 (95%CI 0.76–1.28)	NS HR 1.00 (95%CI 0.78–1.27)	NR	NR	NS
Xu et al. (2022) [7]	Meta-analysis	n = 2412, cStage I	NR	NR	NR	Seg better	NR
Winckelmans et al. (2020) [8]	Meta-analysis	28 studies, n = 8300, cStage I	Comparable for tumors < 2 cm	Comparable for tumors < 2 cm	NR	NR	NR

OS: overall survival, RFS: recurrence-free survival, NS: not significant, NR: not reported, Seg: segmentectomy, Lob: lobectomy, Sub: sublobar.

**Table 2 cancers-17-03988-t002:** Comparison of RATS Segmentectomy Versus VATS and Open Surgery.

Author (Year)	Study Design	Population	OS	RFS	90-Day Mortality	Length of Hospital Stay	Operative Time	Blood Loss	Lymph Node Yield	Complications	Conversion Rate
Kagimoto et al. [11] (2020)	Retro, PSM	n =40 PSM	NR	NR	NS R 0% vs. V 0%	NS: median R 7.5 d vs. V 7.5 d	NS: median R 163.5 m vs. V 147 m	NS: median R 26.5 mL vs. V 33.5 mL	NS	NS R 25% vs. V 20%	NS R 0% vs. V 0%
Zhang et al. [12] (2020)	Retro, PSM	n = 774 (n = 257 PSM)	NR	NR	NS (30-day) R 0% vs. V 0%	NS: median R 4 d vs. V 4 d	NS: average R 147.9 m vs. V 149.2 m	NS: median (ml) R 50 mL vs. V100 mL	R better (LN1)	NS R 17.9% vs. V 14.8%	NS R 0.4% vs. V1.2%
Mao et al. [13] (2021)	Meta-analysis	18 studies, n = 60,349	NR	NR	NS (OR0.72: 95%CI 0.47–1.11)	NS	V better	NR	R better	R better (OR0.85: 95%CI 0.75–0.96)	NS (OR1.42: 95%CI 0.70–2.88)
Montagne et al. [14] (2022)	Retro	n = 174 PSM	NS (3-year) R 90.7% vs. V 82.6%	NS (3-year) R 84.6% vs. V 72.9%	NS R 2.1% vs. V 0.81%	NS: median R 4 d vs. V 4 d	R better: median R 100 m vs. V150 m	NR	NR	NS R 21.1% vs, V32.6%	NS R 2.3% vs. V 10.9%
Gómez-Hernández et al. [15] (2024)	Retro, PSM	n = 204 (n = 146 PSM)	NR	NR	NS (30-Day) R 1.4% vs. V 0%	NS: median R 3 d vs. V 3 d	NS: median R 120 vs. V100	NR	NR	NS R 13.3% vs. V 22.7%	NS R 4% vs, V3.1%
Haruki et al. [16] (2024)	Retro, PSM	n = 231 (n = 126 PSM)	NR	NR	NR	NR	R better: median R 154 m vs. V 210 m	R better: median R 10 mL vs. V 40 mL	NS	NS R 13% vs. V 17%	NR
Pan et al. [17] (2024)	Retro, PSM	n = 594 (n = 225 PSM)	NS (5-year) R 90.5% vs. V 87.9%	NS (5-year) R 83.4% vs. V 83.2%	NR	R better: median R 4 d vs. V 5 d	NS: average R 83.6 m vs. V 80.2 m	R better: median R 10 mL vs. V 40 mL	NS	NS R 20.0% vs. V 26.1%	NS R 2.22% vs, V 1.67%
Caso et al. [18] (2024)	Retro, PSM	n = 22,792 (n = 14,958 PSM)	R, V better (5-year) R 74.1%, V 73.8%, O 69.3%	NR	R, V better R, V 2.5, 2.2% vs. O 4.4%	NR	NR	NR	R better	R, V better (O; higher readmission)	NR
Wang et al. [19] (2024)	Retro	n = 204	NR	NR	NR	R better: median R 4 d vs. V 5 d	R better: average R 58.6 m vs. V 66.1 m	NR	NS	R better	NR
Francis et al. [20] (2024)	Meta-analysis	11 studies, n = 7280	NR	NR	NS; trend V > O > R (R worst)	NR	NR	NR	NR	R better (V, O; higher readmission)	NR
Catelli et al. [21] (2025)	Retro	n = 157	R&V better ( 5-year OS not available)	NR	O higher R 0%, V0% vs. O 6.7%	O longer R 4.9 d, V 5.2 d vs. O 6.3 d	R longer: average R 189 m vs. O&V 153 m	NR	R&O better	NS	R better R 0% vs. V 13%

Retro: retrospective, PSM: propensity score-matched, NS: not significant, NR: not reported, R: robotic-assisted thoracic surgery, V: video-assisted thoracic surgery, O: open thoracic surgery, d: days, m: min.

## Data Availability

The data supporting the findings of this study are available from the corresponding author upon request.

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
