# Peer review of "Refining Surgical Standards: The Role of Robotic-Assisted Segmentectomy in Early-Stage Non-Small-Cell Lung Cancer"

_cancers, 2025, doi:10.3390/cancers17243988_

Round 1

Reviewer 1 Report

Comments and Suggestions for Authors

This review report was aimed to evaluate the benefits of the robotic segmentectomy.

This report is wonderfully organized.

I have several suggestions.

  • Specific figures should be provided in Tables 1 and 2 rather than just NS.
  • Details of the surgical outcomes should be described. For example, the duration of operative time, blood loss, postoperative hospital stays etc in Table 2.
  • There are few descriptions of the benefits of the robotic segmentectomy, and more attention should be paid to other reports, such as RF-ID or VAL-MAP, that describe how to identify tumors in the absence of feedback rather.
  • The CALGB140503 and JCOG0802/WJOG4607L are important trials, but if the main focus is robotic segmentectomy, it would be better to increase the proportion of descriptions of robotic surgery, such as technical innovations and port placements.

I hope this manuscript will be better and publish in the future.

Author Response

Comment 1: Specific figures should be provided in Tables 1 and 2 rather than just NS. Details of the surgical outcomes should be described. For example, the duration of operative time, blood loss, postoperative hospital stays etc in Table 2.

Answer: The authors sincerely thank the reviewer for this valuable comment. In accordance with the suggestion, we have carefully reviewed the data and added specific numerical values to Tables 1 and 2 wherever possible. In addition to the revisions made to the tables, some parts of the main text have also been modified. These textual revisions are highlighted in the manuscript on page 4 (lines 137–144, 165–170, and 185–187), as well as on lines 222–224.

Comment 2: There are few descriptions of the benefits of the robotic segmentectomy, and more attention should be paid to other reports, such as RF-ID or VAL-MAP, that describe how to identify tumors in the absence of feedback rather.

Answer: The authors sincerely thank the reviewer for this important comment. In accordance with the suggestion, we have cited reports indicating the effectiveness of VAL-MAP and RATS for tumor identification in the absence of tactile feedback (lines 361–363). These revisions are highlighted in the manuscript.

Comment 3: The CALGB140503 and JCOG0802/WJOG4607L are important trials, but if the main focus is robotic segmentectomy, it would be better to increase the proportion of descriptions of robotic surgery, such as technical innovations and port placements.

Answer: The authors sincerely thank the reviewer for this important comment. In accordance with the suggestion, we have expanded the discussion in Section 4.3.4 by adding a description of port placement (lines 343–352). These revisions are highlighted in the manuscript

Reviewer 2 Report

Comments and Suggestions for Authors

This is a nice review article summarizing the role of robotic assisted surgical procedures in non-small cell lung cancer treatment.

The authors have made a thorough literature search and found most relevant and up-to-date studies, which they present in the review.
They have included randomized conrtolled trials and have also incorporated international guidelines to compare the utility of robotic segmentectomy (RATS). They have also included comparisons of robotic techniques to video-assisted thorocoscopy and traditional open surgeries.
The tables summarize well the text in the article.

I feel this is an educational review that updates the knowledge on early stage lung cancer surgical treatment techniques.

Author Response

Comment 1: This is a nice review article summarizing the role of robotic assisted surgical procedures in non-small cell lung cancer treatment.

The authors have made a thorough literature search and found most relevant and up-to-date studies, which they present in the review.

They have included randomized conrtolled trials and have also incorporated international guidelines to compare the utility of robotic segmentectomy (RATS). They have also included comparisons of robotic techniques to video-assisted thorocoscopy and traditional open surgeries.

The tables summarize well the text in the article.

I feel this is an educational review that updates the knowledge on early stage lung cancer surgical treatment techniques.

Answer: The authors sincerely thank the reviewer for the positive and encouraging comments regarding our manuscript. We are grateful that the reviewer found the literature search comprehensive and the review educational. We appreciate this feedback and have no further changes in response to this comment.

Reviewer 3 Report

Comments and Suggestions for Authors

The authors reported their work named “Refining Surgical Standards: The Role of Robotic-assisted Segmentectomy in Early-stage Non-small Cell Lung Cancer”. This comprehensive narrative review addresses a timely and clinically significant topic: the paradigm shift from lobectomy to segmentectomy for early-stage NSCLC and the specific role of Robotic-assisted Thoracic Surgery (RATS) in performing these complex resections. The authors effectively synthesize evidence from landmark randomized controlled trials (JCOG0802, CALGB140503) and recent comparative studies to build a compelling case for RATS segmentectomy as a means to balance oncological radicality with functional preservation. The manuscript is well-structured, clearly written, and provides a valuable overview of technical innovations and future directions in the field. I have the following comments:

Major Strengths

  • Topical Relevance: The subject matter is at the forefront of thoracic surgical oncology. The review successfully captures a major contemporary shift in clinical practice.
  • Comprehensive Synthesis: The authors do an excellent job of integrating findings from the two pivotal RCTs (JCOG0802 and CALGB140503), explaining their nuances, differences in design (e.g., segmentectomy-only vs. sublobar resection), and their collective impact on guidelines.
  • Focus on Technical Advancements: The detailed discussion on the technical features of RATS (3D vision, wristed instruments, tremor filtration) and their integration with adjunct technologies like ICG imaging and 3D reconstruction is a major strength. It clearly articulates why RATS is particularly suited for complex anatomical segmentectomies.
  • Future-Oriented Perspective: The inclusion of ongoing trials (STEP-UP, STRONG) and the discussion on AI and personalized surgery provide a forward-looking view that is highly valuable for the reader.

Major Points for Revision

  1. Clarification and Reconciliation of Data in Tables and Text:
    • There are significant discrepancies between the results described in the text and the data summarized in Table 1:

-Text (Page 3): For JCOG0802, it states segmentectomy demonstrated a "statistically significant survival advantage over lobectomy" with 5-year OS of 94.3% vs. 91.1%.

-Table 1: For the same trial, the "OS" and "RFS" columns are marked as "NS" (Not Significant). This is contradictory and misleading. The table must be corrected to accurately reflect the trial's primary findings, perhaps using "Seg better" for OS or providing hazard ratios.

    • Similarly, the text discusses higher local recurrence in JCOG0802, but this critical outcome is not captured in Table 1. Consider adding a "Local Recurrence" column to Table 1 to present this key comparative data.
  1. Critical Analysis of Conflicting Evidence:
    • The results section (3.2) presents comparative data for RATS vs. VATS that is often mixed or conflicting (e.g., on operative time, mortality, complications). The narrative would benefit from a more critical synthesis of these inconsistencies.
    • For example, the text notes that Francis et al. reported "a significantly higher mortality rate in the RATS group," but this critical finding is not discussed in the context of potential confounders (e.g., patient selection, learning curve, center volume). A brief discussion on the potential reasons for these conflicting results (e.g., study design, learning curve effects, patient heterogeneity) is necessary.
  2. Improvement of Table 2 (Comparison of RATS, VATS, Open):
    • Table 2 is currently difficult to interpret. The use of "NS" and "NR" is appropriate, but the directional arrows (e.g., RATS > VATS) are ambiguous without a consistent key. Does RATS > VATS mean better outcome, longer time, higher yield? The reader must constantly refer back to the column header.
    • Recommendation: A legend must be added to explicitly define the symbols. Furthermore, consider using symbols like +, =, - to indicate "better than," "equivalent to," or "worse than" the comparator for outcomes like OS, RFS, and complications, while reserving > and < for quantitative measures like time and length of stay. This would drastically improve readability.

Minor Points

  • Abstract: The final sentence of the abstract is cut off: "...advance". This must be completed.
  • Introduction: The sentence "This review critically examined..." uses past tense. For a review article, present tense is more conventional (e.g., "This review critically examines...").
  • Section 3.2.1: The sentence "However, Catelli et al.[21] reported a 2-year OS of 100% for RATS, 96.2% for VATS, and 75.8% for open surgery." is followed by "Both RATS and VATS demonstrated superior overall survival compared to open surgery." This is correct, but the 100% OS for RATS is an extreme value that may be due to a small sample size or short follow-up. A brief cautionary note would be prudent.

Author Response

Major Comment 1: 

Clarification and Reconciliation of Data in Tables and Text:

There are significant discrepancies between the results described in the text and the data summarized in Table 1:

-Text (Page 3): For JCOG0802, it states segmentectomy demonstrated a "statistically significant survival advantage over lobectomy" with 5-year OS of 94.3% vs. 91.1%.

-Table 1: For the same trial, the "OS" and "RFS" columns are marked as "NS" (Not Significant). This is contradictory and misleading. The table must be corrected to accurately reflect the trial's primary findings, perhaps using "Seg better" for OS or providing hazard ratios.

Similarly, the text discusses higher local recurrence in JCOG0802, but this critical outcome is not captured in Table 1. Consider adding a "Local Recurrence" column to Table 1 to present this key comparative data.

Answer: The authors sincerely thank the reviewer for pointing out these important discrepancies. As suggested, we have corrected the OS entry for JCOG0802 in Table 1 to “Seg better.” In addition, we have added numerical values to Tables 1 and 2 wherever possible. Furthermore, a “Local Recurrence” column has been incorporated into Table 1 to present this critical outcome.

Major Comment 2: 

Critical Analysis of Conflicting Evidence:

The results section (3.2) presents comparative data for RATS vs. VATS that is often mixed or conflicting (e.g., on operative time, mortality, complications). The narrative would benefit from a more critical synthesis of these inconsistencies.

For example, the text notes that Francis et al. reported "a significantly higher mortality rate in the RATS group," but this critical finding is not discussed in the context of potential confounders (e.g., patient selection, learning curve, center volume). A brief discussion on the potential reasons for these conflicting results (e.g., study design, learning curve effects, patient heterogeneity) is necessary.

Answer: The authors sincerely thank the reviewer for this insightful comment. In accordance with the suggestion, we have expanded the discussion in Section 4.1 to provide a more critical synthesis of the inconsistencies between RATS and VATS outcomes. We have also included a brief discussion on possible reasons for conflicting results, such as differences in study design and patient heterogeneity. These revisions are highlighted in the manuscript in Section 4.1 (lines 260–270)

Major Comment 3: 

Improvement of Table 2 (Comparison of RATS, VATS, Open):

Table 2 is currently difficult to interpret. The use of "NS" and "NR" is appropriate, but the directional arrows (e.g., RATS > VATS) are ambiguous without a consistent key. Does RATS > VATS mean better outcome, longer time, higher yield? The reader must constantly refer back to the column header.

Recommendation: A legend must be added to explicitly define the symbols. Furthermore, consider using symbols like +, =, - to indicate "better than," "equivalent to," or "worse than" the comparator for outcomes like OS, RFS, and complications, while reserving > and < for quantitative measures like time and length of stay. This would drastically improve readability.

Answer: 

The authors sincerely thank the reviewer for this valuable comment.

In accordance with the suggestion, we have revised Table 2 to improve clarity. Wherever possible, numerical values have been added to each item, and ambiguous expressions have been corrected.

The authors have also added abbreviations and a legend to explicitly define the symbols used, in order to enhance readability. These revisions are highlighted in the manuscript in Table 2

Minor Comment 1: 

Abstract: The final sentence of the abstract is cut off: "...advance". This must be completed.

Introduction: The sentence "This review critically examined..." uses past tense. For a review article, present tense is more conventional (e.g., "This review critically examines...").

Section 3.2.1: The sentence "However, Catelli et al.[21] reported a 2-year OS of 100% for RATS, 96.2% for VATS, and 75.8% for open surgery." is followed by "Both RATS and VATS demonstrated superior overall survival compared to open surgery." This is correct, but the 100% OS for RATS is an extreme value that may be due to a small sample size or short follow-up. A brief cautionary note would be prudent.

Answer: The authors sincerely thank the reviewer for these helpful comments.

The abstract does not end at the word “advance” but continues on the following page, and we have ensured that the sentence is complete.

In the Introduction, the tense has been revised from past to present in accordance with the reviewer’s suggestion.

Furthermore, in Section 4.1 (page 4, lines 260–270), the authors have added a limitation statement to address the concern regarding the extreme OS value reported by Catelli et al.

These revisions are highlighted in the manuscript.

Reviewer 4 Report

Comments and Suggestions for Authors

Dear Editor and Authors,

It was my pleasure to evaluate this review manuscript titled "Refining Surgical Standards: The Role of Robotic-assisted Segmentectomy in Early-stage Non-small Cell Lung Cancer" by Dr. Masaya Nishino and colleagues from the Division of Thoracic Surgery, Department of Surgery at Kindai University in Osaka, Japan.

In this review the authors presenty the current evidence supporting robotic lung segmentectomy compared to lobectomy for early stage lung cancer.

This is quite a controversial subject in thoracic surgery currently with a number of studies been conducted or under development. For this reviewer the evidence as he has reviewed them and analyzed them is still not robust enough to change his clinical practice which includes lobectomy for all eligible/fit patients and segmentectomy for frail, low respiratory reserves or with multiple co-morbidities patients.

In addition, the comparison between robotic versus thoracoscopic surgery is also still controversial. As both are minimally invasive techniques with similar surgical outcomes the comparisons are down to comfort for the surgeon, operative time and cost. In terms of patient meaningful outcomes such as early recovery, lower complication rates, lower pain ect the two techniques are neck to neck with none really showing a definitive advantage. Of course if you group VATS with open surgery the evidence is against it.

However, this is not what the authors of this work would have you believe. Utilizing certain literature they conclude that robotic segmentectomy has now become the unequivocal "Standard of Care" for early stage NSCLC. Far from it, this is not supported by available literature and clinical practice. 

The higher incidence of locoregional recurrence is also something significant to consider, again for patients which could have undergone a currative lobectomy.

Comments:

  1. A PRISMA diagram needs to be included in this review work.
  2. The number of studies included in the analysis is small and don't encompass all the available literature? Why is this? There seems to be a certain bias towards including studies with certain outcomes.

In conclusion I am not able to support the publication of the work. Thank you.

Author Response

Comment: 

A PRISMA diagram needs to be included in this review work!!

The number of studies included in the analysis is small and don't encompass all the available literature? Why is this? There seems to be a certain bias towards including studies with certain outcomes!!

In conclusion I am not able to support the publication of the work. Thank you.

Answer: 

The authors sincerely appreciate the reviewer for the critical and thoughtful comments.

As this manuscript is a narrative review, a PRISMA diagram is not necesary and therefore was not included.

The authors acknowledge that narrative reviews may inherently contain more bias compared to systematic reviews, and kindly ask for the reviewer’s understanding in this regard.

Nevertheless, receiving such critical and sincere feedback is extremely valuable for us, and we consider it an important opportunity for growth, including in the refinement of our academic.

Round 2

Reviewer 1 Report

Comments and Suggestions for Authors

Thank you for the revisions, and the revised manuscript deserves acceptance.

Author Response

Comment1: Thank you for the revisions, and the revised manuscript deserves acceptance.

Answer: The authors sincerely thank the reviewer for the constructive comments and for the cooperation that has led to the acceptance of this manuscript.

Reviewer 3 Report

Comments and Suggestions for Authors

The authors addressed my comments and I am pleased to accept their work.

Author Response

Comment1: The authors addressed my comments and I am pleased to accept their work.

Answer: The authors sincerely appreciate the reviewer’s thoughtful evaluation and are grateful for the decision to accept our work.

Reviewer 4 Report

Comments and Suggestions for Authors

Dear Editor and Authors,

I read and re-evaluated the revised manuscript. Although my minor suggestions have been implemented my major one which is the bias towards RATS versus VATS has not been adequately addressed. The authors have made some minor changes trying to offer a more ballanced approach but still the paper reads too much in favour of RATS as opposed to minimally invasive surgery vs conventional. 

As previously mentioned there are no solid evidence to suggest RATS superiority over VATS other than ergonomics for the surgeon!! I therefore remain reluctant to recommend its publication. Therefore, I suggest that the authors re-revisit their work and tone down their "enthusiasm" for RATS!!

Kind regards to all and I am awaiting a revised version of this work.

Author Response

Comment1: I read and re-evaluated the revised manuscript. Although my minor suggestions have been implemented my major one which is the bias towards RATS versus VATS has not been adequately addressed. The authors have made some minor changes trying to offer a more ballanced approach but still the paper reads too much in favour of RATS as opposed to minimally invasive surgery vs conventional.

As previously mentioned, there are no solid evidence to suggest RATS superiority over VATS other than ergonomics for the surgeon!! I therefore remain reluctant to recommend its publication. Therefore, I suggest that the authors re-revisit their work and tone down their "enthusiasm" for RATS!!

Kind regards to all and I am awaiting a revised version of this work.

Answer: The authors sincerely appreciate for the reviewer’s valuable comments and acknowledge that our initial draft may have overstated the superiority of RATS despite the limited solid evidence currently available. In response, we have made several revisions to address your concerns:

  • In Discussion section 4.4, we have deleted the subsection entitled “The technical features of RATS directly translate into better clinical outcomes” (lines 394–405).
  • We have added a new Limitation paragraph at the end of section 4.4 to emphasize the restricted evidence base and the need for further studies.
  • In the Conclusion, the first and second paragraphs have been revised to avoid overemphasizing the advantages of RATS, ensuring a more balanced tone.
  • In Discussion section 4.3, we have also refined the topic presentation to improve clarity and consistency.

All changes have been highlighted in the revised manuscript for your convenience.

Round 3

Reviewer 4 Report

Comments and Suggestions for Authors

Dear Editor and Authors,

I read and evaluated once more the revised manuscript submitted by the authors. They have now implemented a more ballanced approach as asked and therefore I am happy to now recommend its publication.

Kind regards.